# OpenReview forum: "Teach to Reason Safely: Policy-Guided Safety Tuning for MLRMs"
_ICLR.cc/2026/Conference — ICLR 2026 Poster_

### Official Review · Reviewer_c61W · 2025-10-28

**Soundness:** 2
**Presentation:** 3
**Contribution:** 2
**Rating:** 4
**Confidence:** 4

**Summary:**

This paper investigates the safety-reasoning trade-off in Multimodal Large Reasoning Models (MLRMs), highlighting that improved reasoning often coincides with increased harmful output rates. The authors identify two key mechanisms behind safety degradation—visual attention drift and unsafe reasoning patterns. To address these, they propose Policy-Guided Safety Tuning (PST), a two-stage alignment framework consisting of Policy-Guided Supervised Fine-Tuning (PST-SFT) and Safety Reasoning Preference Optimization (SRPO). Experiments on multiple benchmarks show that PST significantly reduces harmful outputs without major sacrifices in general reasoning capabilities.

**Strengths:**

1. The paper offers a clear and organized examination of safety degradation in MLRMs, identifying two representative mechanisms—visual attention drift and unsafe reasoning patterns. While parts of the analysis are qualitative, the work provides a useful foundation for understanding how reasoning-oriented tuning may influence safety performance across benchmarks.
2. The proposed two-stage PST framework effectively integrates explicit safety policies into reasoning, moving beyond refusal-based alignment toward interpretable, reasoning-centered safety control.
3. The dataset construction is detailed and replicable, with Figure 6 making the process and logic transparent.

**Weaknesses:**

1. The motivation underlying the claimed "reasoning–safety trade-off" is not entirely sound. In Section 3.1, the authors compare reasoning-tuned models with their corresponding base models and attribute the observed safety degradation to reasoning itself. However, this conclusion lacks causal rigor. The performance gap may simply stem from the absence of safety-aware data during reasoning-tuning rather than an inherent conflict between reasoning and safety. Moreover, the analysis does not control for data composition or fine-tuning objectives, which weakens the validity of the stated correlation.
2. The failure analysis in Section 3.2 appears rather subjective. The authors summarize two types of failure modes but do not describe the process or key data used to derive them. For example, it remains unclear what proportion of the failed cases fall under *Visual Attention Drift* or *Unsafe Reasoning Patterns*. Moreover, no empirical evidence is provided to show that *Visual Attention Drift* directly causes unsafe behavior, the logical connection between the two is not clearly established, even though they are observed to co-occur statistically.
3. The proposed Policy-Guided Safety Tuning (PST) framework is conceptually similar to existing approaches (e.g., Sun et al., 2023; Guan et al., 2024; Wang et al., 2025). The paper does not clearly articulate how PST substantively differs from or advances beyond these prior frameworks beyond its multimodal setting.

[1] Sun Z, Shen Y, Zhou Q, et al. Principle-driven self-alignment of language models from scratch with minimal human supervision[J]. Advances in Neural Information Processing Systems, 2023, 36: 2511-2565.

[2] Guan M Y, Joglekar M, Wallace E, et al. Deliberative alignment: Reasoning enables safer language models[J]. arXiv preprint arXiv:2412.16339, 2024.

[3] Wang H, Qin Z, Shen L, et al. Safety Reasoning with Guidelines[C]//Forty-second International Conference on Machine Learning.

**Questions:**

1. The SRPO objective (Eq. 7) appears mathematically identical to standard DPO, with the same log-ratio formulation and sigmoid preference loss. Since both $y_w$ and $y_l$ already include reasoning traces, could the authors clarify what differentiates SRPO conceptually or algorithmically from conventional DPO?

---

> ### Author Response · Authors · 2025-11-27
> **Response to Reviewer c61W**
>
> ### W1: The motivation underlying the claimed "reasoning–safety trade-off" is not entirely sound.
>
> We thank the reviewer for the insightful comment. Regarding the concern that the observed performance gap might stem from the lack of safety-aware signals during reasoning tuning rather than reflecting a fundamental conflict between reasoning and safety, we provide the following clarifications.
>
> To evaluate the effect of explicit reasoning, we compared two variants of the **same model** in Section 3.1: one producing multi-step reasoning traces (with CoT) and the other generating direct answers (w/o CoT). The results, shown in the table below and in Figure 2 of the manuscript, indicate that the variants with explicit reasoning consistently exhibit higher harmful response rates (HR) across all evaluated architectures. This suggests that generating multi-step reasoning inherently increases the likelihood of producing harmful responses.
>
> | **Model (Method)**       | **Beavertails** | **SPA-VL** |
> | ------------------------ | --------------- | ---------- |
> | OceanR1 (w/o CoT)        | 57.72           | 15.45      |
> | OceanR1 (w CoT)          | 81.15           | 63.02      |
> | Mulberry-LLaVA (w/o CoT) | 76.74           | 43.77      |
> | Mulberry-LLaVA (w CoT)   | 83.87           | 62.26      |
> | Mulberry-LLaMA (w/o CoT) | 41.77           | 21.51      |
> | Mulberry-LLaMA (w CoT)   | 86.25           | 62.64      |
> | MMEureka (w/o CoT)       | 65.53           | 30.94      |
> | MMEureka (with CoT)      | 74.02           | 41.13      |
>
> Prior work further provides both theoretical and empirical support for this interpretation. Safemlrm [1] shows that longer reasoning chains tend to increase a model’s susceptibility to harmful goal-directed cues, as extended reasoning paths create additional opportunities for failure propagation and unsafe completion. Think in Safety [2] demonstrates that reasoning-tuned models systematically exhibit safety alignment collapse under adversarial prompts, and identifies CoT trajectories as key amplifiers of unsafe behavior unless targeted safety interventions are applied.
>
> Taken together, these findings suggest that the observed phenomenon is not merely due to the absence of safety-aware signals during reasoning tuning, but reflects a deeper interaction between explicit reasoning processes and adversarial vulnerability.
>
> We thank the reviewer again for raising this important point and will revise the manuscript to more clearly convey the scope and implications of our findings.
>
> [1] SafeChain: Safety of Language Models with Long Chain-of-Thought Reasoning Capabilities, in ACL 2025.
>
> [2] Think in Safety: Unveiling and Mitigating Safety Alignment Collapse in Multimodal Large Reasoning Model, in EMNLP 2025.

---

> ### Author Response · Authors · 2025-11-27
> **Response to Reviewer c61W**
>
> ### W2: The failure analysis in Section 3.2 appears rather subjective.
>
> Thank you for your comment regarding the subjectivity of our failure analysis in Section 3.2. To clarify, we analyzed failure cases from both perception and reasoning perspectives, identifying common behaviors across models that led us to categorize failures as Visual Attention Drift or Unsafe Reasoning Patterns (termed in Section 3.2 as *Flawed Reasoning Initiation / Chain-of-Thought Safety Attenuation*).
>
> To provide quantitative evidence, we counted the number of failures corresponding to these categories across four datasets: BeaverTails-V (589), SPA-VL (256), MM-SafetyBench (1680), and SIUO (167). Each entry in the Table reports the number of failures for **Visual Attention Drift**, **Flawed Reasoning Initiation**, and **Chain-of-Thought Safety Attenuation**, respectively.
>
> | Model            | BeaverTails-V (589) | SPA-VL (256)  | MM-SafetyBench (1680) | SIUO (167)   |
> | ---------------- | ------------------- | ------------- | --------------------- | ------------ |
> | R1-Onevision     | 57 / 331 / 88       | 24 / 118 / 24 | 165 / 429 / 119       | 32 / 62 / 24 |
> | R1-Onevision+PST | 19 / 27 / 30        | 2 / 6 / 4     | 19 / 30 / 29          | 5 / 11 / 5   |
> | LLaMA-CoT        | 64 / 336 / 91       | 28 / 95 / 39  | 198 / 417 / 105       | 13 / 54 / 11 |
> | LLaMA-CoT+PST    | 16 / 37 / 39        | 2 / 5 / 6     | 26 / 32 / 27          | 9 / 16 / 4   |
>
> Baseline models (R1-Onevision and LLaMA-CoT) exhibit substantial failures in both perception- and reasoning-related categories. Applying PST significantly reduces failures across all categories and datasets—for example, R1-Onevision on BeaverTails-V decreases from 57 / 331 / 88 to 19 / 27 / 30—demonstrating that PST effectively mitigates unsafe behaviors arising from both attention drift and reasoning errors.
>
> -----------------
>
> Regarding Visual Attention Drift, we analyzed attention distributions across tokens in reasoning and non-reasoning models. As shown in Figure 3, reasoning-based models assign lower weights to visual tokens, particularly in deeper layers.
>
> To further investigate the effect of visual attention on model safety, we conducted an experiment comparing two inference settings: (1) generating a caption before reasoning, and (2) direct reasoning. The results, summarized in Table Y, show that forcing the model to attend to visual content via caption generation reduces harmlessness performance (HR), indicating that inappropriate visual attention can directly contribute to unsafe behavior.
>
>
>
> | Model                      | BeaverTails-V | SPA-VL | MM-SafetyBench | SIUO  |
> | -------------------------- | ------------- | ------ | -------------- | ----- |
> | R1-Onevision (w/o caption) | 78.61         | 52.83  | 30.89          | 83.83 |
> | R1-Onevision (w caption)   | 75.72         | 50.78  | 29.88          | 77.84 |
> | LLaMA-CoT (w/o caption)    | 83.87         | 82.26  | 33.93          | 73.65 |
> | LLaMA-CoT (w caption)      | 79.96         | 80.46  | 33.09          | 73.05 |

---

> ### Author Response · Authors · 2025-11-27
> **Response to Reviewer c61W**
>
> ### W3: The paper does not clearly articulate how PST substantively differs from or advances beyond these prior frameworks beyond its multimodal setting
>
> Thank you for the comment. Previous methods primarily focused on injecting safety policies, which only address the problem of Flawed Reasoning Initiation—that is, ensuring the model references or assesses relevant safety policies before analyzing a problem. PST introduces several distinct innovations, particularly in the context of multimodal safety-aware reasoning:
>
> 1. Innovative Problem Definition and Data Construction
>    We systematically define and operationalize challenging safety-related failure modes, including Visual Attention Drift and Unsafe Reasoning Patterns, and design targeted templates and experiments to capture these risks at a fine-grained level. This structured problem formulation enables precise identification and mitigation of model vulnerabilities.
> 2. Operationalization in Model Training
>
> During data preparation, images are converted into textual captions and aligned with questions, ensuring that the model receives accurate and semantically rich visual representations prior to reasoning.In the PST-SFT stage, the model adopts a “Description First” format, generating a caption describing key image content before answering questions. This provides a stable semantic anchor, mitigating mispredictions caused by attention drift.
>
> To address Chain-of-Thought Safety Attenuation, our innovation lies in the careful construction of negative preference samples. Through Comparative Failure Mining, we deliberately design instances that begin with an initially correct and safe analysis but gradually deviate, ultimately leading to an incorrect conclusion. By treating these “seemingly correct but ultimately flawed” examples as negative samples, the SRPO stage teaches the model not only to initiate reasoning correctly but also to maintain safe reasoning logic throughout, providing deep reinforcement of safety robustness.
>
> In summary, PST introduces a systematic, multimodal approach to safety-aware reasoning by combining fine-grained problem definition with operationalized training strategies. Collectively, these innovations extend beyond prior work by providing both concrete guidance and empirical validation for robust safety alignment in multimodal reasoning models.
>
> ### Q1: could the authors clarify what differentiates SRPO conceptually or algorithmically from conventional DPO?
>
> Thank you for the comment. The key difference lies in the data and preference modeling. SRPO is specifically designed for safety-aware reasoning tasks and uses datasets that explicitly include Chain-of-Thought (CoT) reasoning trajectories. These data contain not only the final answers but also detailed intermediate reasoning steps. In contrast, standard DPO primarily models preferences over final outputs.

---

### Official Review · Reviewer_vzoQ · 2025-10-28

**Soundness:** 3
**Presentation:** 3
**Contribution:** 2
**Rating:** 6
**Confidence:** 4

**Summary:**

The paper
- Illustrates that reasoning VLMs generate harmful outputs at a higher rate than their non-reasoning counterparts.
- Introduces a multi-step data generation pipeline that generates policy-specific reasoning traces on the basis of text-image samples from the Beavertails-V dataset.
- The data generation pipeline produces both SFT as well as preference data.
- Fine-tunes existing reasoning VLMs on the generated data and evaluates the results on a range of safety and general vision-language reasoning benchmarks

**Strengths:**

- Extensive experiments across multiple benchmarks show improved safety performance for PST fine-tuned models while maintaining high general Vision Language reasoning capabilities.
- The method seems to outperform recent comparable methods such as MSR-align and Think-in-Safety.

**Weaknesses:**

- Paper claims that reduced attention to visual tokens is cause of safety degradation in reasoning models, but it doesn’t show whether the PST fine-tuning actually changes that behavior.
- The conceptual contribution is fairly limited as the paper follows a well-established pattern of using a multistep pipeline with powerful third-party models to curate a dataset that enables distilling of the in-context capabilities of these powerful models into smaller models and improve their performance on respective benchmarks.

**Questions:**

- The data generation process between MSR-align, Think-in-Safety and the proposed method are all very similar. It is very difficult to understand what the main reason is for PST to outperform these prior works. Is it a different choice of prompts in the pipeline? A different choice of models? Specific steps in the pipeline?

---

> ### Author Response · Authors · 2025-11-27
> **Response to Reviewer vzoQ**
>
> ### W1: It doesn’t show whether the PST fine-tuning actually changes reduced attention to visual token behavior.
>
> To verify whether PST fine-tuning can mitigate failures caused by visual attention drift, we counted the number of failures attributable to attention drift for each model across different datasets—that is, cases where the model failed to focus on key visual information, resulting in incorrect answers. The results are shown in the table below:
>
> |                  | BeaverTails-V | SPA-VL | MM-SafetyBench | SIUO |
> | ---------------- | ------------- | ------ | -------------- | ---- |
> | R1-Onevision     | 57            | 24     | 165            | 32   |
> | R1-Onevision+PST | 19            | 2      | 19             | 5    |
> | LLaMA-CoT        | 64            | 28     | 198            | 13   |
> | LLaMA-CoT+PST    | 16            | 2      | 26             | 9    |
>
> It can be observed that PST fine-tuning significantly reduces failures caused by attention drift. For example, R1-Onevision’s failures on BeaverTails-V decrease from 57 to 19, while LLaMA-CoT’s failures on MM-SafetyBench drop from 198 to 26.
>
> Furthermore, as illustrated in the case comparisons in the appendix (particularly Case 4), by continuously referencing visual information, PST fine-tuning can maintain safe and compliant reasoning while producing accurate answers, effectively mitigating the issue of attention drift.
>
>
>
> ### W2: Multi step distillation pipeline.
>
> While our approach leverages established distillation techniques, the primary conceptual contribution of this work goes beyond merely executing a multi-step distillation pipeline. This paper addresses a previously key problem in Multimodal Large Reasoning Models (MLRMs): the safety-reasoning trade-off. We identify specific mechanisms underlying safety degradation and propose targeted interventions, which we believe constitute the key conceptual novelty.
>
> 1. **Identifying and Analyzing Safety Degradation Mechanisms**
>
> We conduct a large-scale analysis revealing two distinct mechanisms that contribute to safety degradation during reasoning-oriented fine-tuning:
>
>  **Visual Attention Drift:** Fine-tuning for reasoning can reduce the model’s reliance on visual grounding, causing it to overlook potentially harmful visual content and rely instead on “text shortcuts.”
>
>  **Unsafe Reasoning Patterns:** These include Flawed Reasoning Initiation (e.g., self-rationalization that bypasses safety constraints) and Chain-of-Thought Safety Attenuation, where safety awareness progressively diminishes over multi-step reasoning.
>
> These mechanisms have not been adequately addressed in prior work, and our systematic characterization provides a foundation for designing interventions to mitigate unsafe behavior.
>
> 2. **Policy-Guided Safety Alignment Framework (PST)**
>
> The key conceptual novelty of PST lies not in inventing a new distillation algorithm, but in upstream, foundational interventions that guide models toward safe reasoning:
>
> **Data Construction with Rich Visual Semantics:** Images are converted into textual captions and aligned with questions, ensuring the model receives accurate and semantically rich visual representations prior to reasoning. During the PST-SFT stage, the model first generates a caption describing key image content before answering questions, providing a stable semantic anchor that mitigates mispredictions caused by attention drift.
>
>  **Suppressing Flawed Reasoning Initiation:** The model is required to reference or assess relevant safety policies before proceeding with analysis. This embeds a “safety-first reflex,” ensuring both safe outputs and interpretable reasoning.
>
>  **Suppressing Chain-of-Thought Safety Attenuation:** Using Comparative Failure Mining, we construct negative preference examples that begin with correct reasoning but gradually deviate toward unsafe conclusions. Through the SRPO stage, the model learns to maintain safe reasoning logic throughout multi-step reasoning, reinforcing long-term safety robustness.

---

> ### Author Response · Authors · 2025-11-27
> **Response to Reviewer vzoQ**
>
> ###  Q1: Main reason is for PST to outperform these prior works.
>
> Unlike existing methods that rely solely on SFT , such as MSR-align and Think-in-Safety, our approach incorporates both an SFT stage and an additional SRPO stage.
>
> In the SFT stage, the model primarily learns to imitate high-quality responses. In contrast, the SRPO stage focuses on teaching the model to distinguish relative quality—i.e., to select the superior and safer answer when presented with two superficially reasonable responses. Through this preference optimization, the model not only learns “what constitutes a good answer” but also understands “why one answer is better than another,” thereby achieving a higher level of alignment. Moreover, SRPO enhances safety while better preserving the model’s general capabilities.
>
> During data construction, we further introduce Comparative Failure Mining and Post-hoc Adversarial Reasoning Generation. These examples begin with seemingly correct reasoning but gradually deviate toward unsafe conclusions. Through the SRPO stage, the model learns not only to initiate reasoning correctly but also to maintain safe reasoning logic throughout, providing a deeper reinforcement of safety robustness.
>
> Compared with existing methods that primarily rely on limited-dimension safety filtering, our approach employs multi-dimensional quality evaluation and filtering. Specifically, each generated sample is assessed across multiple dimensions—including Safety, Policy Relevance, Logical Accuracy, Multimodal Coherence, and Helpfulness—through both automated and human evaluation, ensuring high-quality supervision.

---

### Official Review · Reviewer_ZsCz · 2025-10-29

**Soundness:** 1
**Presentation:** 2
**Contribution:** 2
**Rating:** 2
**Confidence:** 4

**Summary:**

The paper studies a safety–reasoning trade-off in Multimodal Large Reasoning Models (MLRMs): reasoning-tuned models produce more harmful outputs than their base counterparts. It analyzes two mechanisms—(i) visual attention drift that weakens visual grounding and (ii) unsafe reasoning patterns such as flawed reasoning initiation and chain-of-thought safety attenuation. The authors also propose Policy-guided Safety Tuning (PST), a two-stage framework: Policy-Guided Supervised Fine-Tuning (PST-SFT) that embeds explicit safety policies into reasoning trajectories, and Safety Reasoning Preference Optimization (SRPO) that aligns toward safe yet informative responses via a preference loss. Experiments on BeaverTails-V, MM-SafetyBench, SPA-VL, SIUO, and general VL benchmarks show reduced harmful rate (HR) and refusal rate (RR) while maintaining competitive VQA/MathVista performance.

**Strengths:**

1. The paper addresses a timely problem: how to teach reasoning models to think safety.
2. Experiments indicate that PST attains reasonable safety gains without severely harming general capabilities.
3. The paper is well written.

**Weaknesses:**

1. **Limited depth of insight.** One of the paper’s central mechanisms, **Visual Attention Drift**, appears to be directly taken from [1] rather than newly discovered or substantially extended in this work.
2. **Motivation and experiments are not well aligned.** The Introduction (Section 1) and the Analysis (Section 3) devote substantial space to Visual Attention Drift and Unsafe Reasoning Patterns, yet the experiments do not demonstrate how PST resolves or alleviates these two phenomena. I did not find corresponding analyses in the main text or the appendix.
3. **Method novelty is incremental.** The two key components, **PST-SFT** and **SRPO**, are essentially direct transfers of widely used techniques (**SFT** and **DPO**). The overall **SFT + DPO** training pipeline is now mainstream; thus, much of the observed improvement likely stems from established methods rather than a fundamentally new algorithmic contribution.
4. **Data and reproducibility.** Following 3, the main innovation appears to lie in data processing and dataset construction. Building the dataset depends on additional models (e.g., Qwen-vl, DeepSeek-r1, GPT-4o), yet the costs in time, storage, and token usage are unclear, and it is not stated whether the data or code will be released.

[1] Chengzhi Liu, Zhongxing Xu, Qingyue Wei, Juncheng Wu, James Zou, Xin Eric Wang, Yuyin Zhou, and Sheng Liu. More thinking, less seeing? assessing amplified hallucination in multimodal reasoning models. arXiv preprint arXiv:2505.21523, 2025a.

**Questions:**

1. How, quantitatively, does PST reduce Visual Attention Drift, and how do before–after attention changes correlate with harmful-rate reductions?
2. Do Unsafe Reasoning Patterns decrease after PST, and how are these patterns operationalized and measured across reasoning steps?
3. Beyond standard SFT and DPO, what are the specific algorithmic novelties of PST-SFT and SRPO, and how sensitive are gains to these components?
4. What are the token, time, compute, and storage costs of constructing DSFT and DSRPO?

---

> ### Author Response · Authors · 2025-11-27
> **Response to Reviewer ZsCz**
>
> ### W1 & Q1: Limited Depth of Insight
>
> Thank you for constructive comments. The contribution of this paper does not lie in "discovering" the phenomenon of visual attention drift itself. For the first time, we re-examine this phenomenon from the novel and crucial perspective of AI safety, revealing its underlying mechanism as a key security risk. Furthermore, we propose a mitigation approach through the generation of captions during the reasoning process, with a clear reference to [1] in the manuscript.
>
> [1] Chengzhi Liu, Zhongxing Xu, Qingyue Wei, Juncheng Wu, James Zou, Xin Eric Wang, Yuyin Zhou, and Sheng Liu. More thinking, less seeing? assessing amplified hallucination in multimodal reasoning models. arXiv preprint arXiv:2505.21523, 2025a.
>
> ### W2 & Q2: Motivation and Experiments
>
> ### 1) Mitigating  Visual Attention Drift
>
> Our approach to mitigating visual attention drift begins at the stage of data construction, where we ensure modality alignment by converting image content into textual captions, which are then paired with subsequent questions. This design guarantees that the model obtains reliable and accurate semantic representations before performing reasoning, thereby substantially reducing the misleading effects of attention drift.
>
> During the PST-SFT phase, the model adopts a "Description First" response format throughout the reasoning process. Specifically, before answering any question, the model is required to first generate a textual description of the key content in the image, thus providing a robust semantic anchor for subsequent reasoning. This strategy serves to markedly reduce misjudgments arising from attention drift.
>
> Representative cases illustrating this design philosophy can be found in Appendix E.1 of the manuscript. These examples demonstrate the model's "description-before-reasoning" trajectory, ensuring accurate comprehension of visual information.
>
> ### 2) Addressing Unsafe Reasoning Patterns
>
> To address flawed reasoning initiation, we explicitly require the model to reference or assess relevant safety policies prior to engaging in question analysis. Only upon confirming that the inquiry falls within safe and compliant boundaries does the model proceed to further reasoning. This strategy is intended to instill a "safety-first" reflex arc within the model, ensuring that the generated answers are not only safe in their conclusions, but also adhere to compliant and interpretable reasoning processes. This approach effectively reinforces the model’s safety boundaries.
>
> To suppress the phenomenon of chain-of-thought safety attenuation, we further innovate in the careful construction of preference optimization negative samples. Through Comparative Failure Mining, we deliberately design initial erroneous responses whereby the model may begin with a correct and safe analysis, but subsequently veers off course, ultimately arriving at an incorrect conclusion. By using such “apparently correct but ultimately flawed” samples as negatives during suppression, our SRPO phase trains the model not only to initiate responses safely, but also to consistently maintain safe reasoning throughout. This constitutes a significant reinforcement of the model’s robustness in safety.
>
> To provide quantitative evidence, we counted the number of failures in each category across four datasets—BeaverTails-V, SPA-VL, MM-SafetyBench, and SIUO. The table reports the number of failures corresponding to Visual Attention Drift, Flawed Reasoning Initiation, and Chain-of-Thought Safety Attenuation, respectively.
>
> |                  | BeaverTails-V (589) | SPA-VL (256)  | MM-SafetyBench (1680) | SIUO (167)   |
> | ---------------- | ------------------- | ------------- | --------------------- | ------------ |
> | R1-Onevision     | 57 / 331 / 88       | 24 / 118 / 24 | 165 / 429 / 119       | 32 / 62 / 24 |
> | R1-Onevision+PST | 19 / 27 / 30        | 2 / 6 / 4     | 19 / 30 / 29          | 5 / 11 / 5   |
> | LLaMA-CoT        | 64 / 336 / 91       | 28 / 95 / 39  | 198 / 417 / 105       | 13 / 54 / 11 |
> | LLaMA-CoT+PST    | 16 / 37 / 39        | 2 / 5 / 6     | 26 / 32 / 27          | 9 / 16 / 4   |
>
> PST fine-tuning substantially reduces failures in both perception- and reasoning-related categories, demonstrating effective mitigation of unsafe behaviors.

---

> > ### Author Response · Authors · 2025-11-27
> > **Response to Reviewer ZsCz**
> >
> > ## W3 & Q3: Method Novelty is Incremental
> >
> > We wish to clarify that the core contribution of this work is not to propose a fundamentally new algorithmic alternative to SFT or DPO. Rather, our innovation is both problem-driven and data-centric. Specifically, our principal contributions are:
> >
> > 1. **Systematic Identification of Critical Failure Modes:** We are the first to systematically identify and define a set of previously overlooked, high-impact failure modes in multimodal safety, including visual attention drift and safety guardrail decay.
> > 2. **Data-Centric Intervention:** We develop a novel and sophisticated data construction methodology that enables mainstream training strategies, such as SFT and DPO, to function as targeted instruments for addressing these issues.
> >
> > The majority of observed performance improvements are not solely due to the generic capabilities of SFT and DPO, but are directly attributable to our identification of key problems and our tailored data-driven solutions.
> >
> > Prior to our work, safety-related failures in multimodal models were often described vaguely, e.g., as “unsafe” or “incorrect reasoning.” Our first major contribution is to distill these ambiguous phenomena into concrete, actionable failure patterns:
> >
> > - **Visual Attention Drift:** The model’s ability to incorporate critical visual cues necessary for safe reasoning is weakened, increasing reliance on linguistic priors and leading to “text shortcuts.” In such cases, responses are driven primarily by textual cues rather than a comprehensive multimodal understanding.
> >
> >   **Flawed Reasoning Initiation:** Following the prevailing “reasoning-first” training paradigm, models often fail to safely handle harmful requests during initial reasoning steps. This manifests in two ways:
> >
> >   1. **Self-rationalization:** The model reframes harmful instructions as benign (e.g., presenting them as a “hypothetical scenario”), circumventing established safety guardrails.
> >
> >   2. **Task-driven cognitive tunneling:** The model becomes overly focused on completing subtasks, neglecting broader safety considerations.
> >
> >  In both cases, the flawed initial premise establishes a foundation that propagates unsafe reasoning throughout the response.
> >
> >   **Chain-of-Thought Safety Attenuation:** Unsafe outcomes can also result from gradual erosion of safety during extended reasoning. Even when reasoning begins with appropriate safety constraints, they may weaken as the chain unfolds. Minor deviations accumulate, leading to conclusions that violate safety policies despite initial compliance.
> >
> > By introducing this precise analytical framework and evaluation targets, we provide the field with an important conceptual advance.
> >
> > - Our PST-SFT and SRPO protocols are not mere adaptations of conventional SFT or DPO. Rather, they systematically incorporate our nuanced understanding of the identified failure modes into every stage of the data construction and training process:
> >
> >   **PST-SFT**: Beyond standard fine-tuning, PST-SFT serves as behavioral scaffolding. Instead of relying on generic instruction data, we employ structured templates based on “description-first” and “policy-priming” paradigms. This design ensures that the model acquires correct behavioral heuristics during fine-tuning, explicitly targeting and mitigating specific failure modes, in contrast to conventional SFT, which primarily aims to improve general instruction-following.
> >
> >   **SRPO**: In contrast to traditional DPO, which typically utilizes clearly incorrect negative samples, our approach focuses on high-quality adversarial negatives. These rejected samples begin with accurate and safe reasoning but progressively deviate toward unsafe or incorrect conclusions. Training on such examples teaches the model not only to initiate reasoning correctly but also to maintain safety and consistency across extended reasoning chains. These adversarial “game-theoretic” samples provide richer and more informative supervision signals than conventional binary preference pairs. While DPO functions as the optimization algorithm, the carefully constructed challenging data supplies the critical learning signals necessary to achieve robust and safe reasoning behavior.
> >
> > ## W4 & Q4: Data and Reproducibility
> >
> > We appreciate the reviewer’s comments. Following publication of this paper, we will publicly release all code used for data generation, processing, and evaluation. This includes scripts for dataset filtering, the prompting logic for our LLM-as-a-Judge methodology, and the algorithms.
> >
> > For data generation, we deployed local models such as DeepSeek and Qwen-VL on a server equipped with eight NVIDIA A100 80GB GPUs. The total computational time for data processing was approximately 50 GPU hours. During the final preference annotation and voting stage, we utilized the GPT-4o API, with a total consumption of roughly 5 million input tokens and 8 million output tokens.

---

> > > ### Comment · Reviewer_ZsCz · 2025-11-28
> > > **Response to authors**
> > >
> > > Thank you for providing detailed clarifications in your rebuttal, especially regarding the methodological contributions and experimental results. This addresses most of my concerns. Please ensure that these clarifications are included in the final manuscript.

---

> > > > ### Author Response · Authors · 2025-11-29
> > > > **Official Comment by Authors to Reviewer ZsCz**
> > > >
> > > > Dear Reviewer ZsCz:
> > > >
> > > > Thank you for the response and the recognition of our rebuttal. We are glad that our clarifications have addressed most of your concerns. We briefly restate our core contributions and will incorporate all expanded explanations and experimental results into the final manuscript.
> > > >
> > > > Our work provides a systematic identification of three critical failure modes in the safety of multimodal large reasoning models: Visual Attention Drift, Flawed Reasoning Initiation, and Chain-of-Thought Safety Attenuation. Based on these findings, we introduce two targeted solutions, PST-SFT and SRPO. These approaches directly address the identified issues rather than offering incremental extensions of existing methods. The quantitative results show that PST substantially reduces perception-related and reasoning-related safety failures across all evaluated benchmarks.
> > > >
> > > > We also commit to releasing all code and data to ensure reproducibility.
> > > >
> > > > We will revise the manuscript accordingly to present these contributions more clearly. We believe the strengthened presentation highlights the novelty and practical impact of our work.
> > > >
> > > > Best regards,
> > > >
> > > > Authors

---

### Official Review · Reviewer_4x5E · 2025-10-31

**Soundness:** 3
**Presentation:** 3
**Contribution:** 2
**Rating:** 6
**Confidence:** 4

**Summary:**

The authors present evidence that multi-modal reasoning models significantly outperform multi-modal base models on complex tasks; however, this improvement also leads to an increase in the generation of harmful content. Through experimental analysis, they identify two primary causes: 1) visual attention drift and 2) unsafe reasoning patterns. They note that existing methods primarily focus on teaching models how to reject harmful outputs without providing guidance on safe reasoning. To address this issue, the authors propose a two-stage alignment framework called Policy-guided Safety Tuning. Testing on various multi-modal safety benchmarks demonstrates that this method significantly reduces the rate of harmful content generation while also performing well on Visual Question Answering tasks, without exhibiting issues of over-sensitivity.

**Strengths:**

1. The paper is well-written, demonstrating clarity and coherence throughout.
2. The authors provide a thorough comparison of multiple baseline methods, testing their proposed approach against a diverse array of benchmarks.
3. The paper explores the relationship between multimodal attention mechanisms and safety considerations, contributing valuable insights to the field.
4. The motivation is really good.

**Weaknesses:**

The proposed method heavily relies on the quality of the training data, which is entirely AI-generated and evaluated. This might introduce a significant bias.

**Questions:**

1. It might be better if the authors clarity their contribution as a dataset.
2. Could you explain how the paper addresses potential biases introduced by using AI-generated data, such as manual annotation or other approaches for data quality control?
3. JailbreakV [1] shows that even randomly generated images and harmful questions could attack MLLMs easily. Could the authors explain the effectiveness of PST under these conditions? Additionally, how do unrelated image descriptions impact the model's responses?
[1] JailBreakV: A Benchmark for Assessing the Robustness of MultiModal Large Language Models against Jailbreak Attacks

---

> ### Author Response · Authors · 2025-11-27
> **Response to Reviewer 4x5E**
>
> ### W1: It might be better if the authors clarity their contribution as a dataset.
>
> Thanks to the reviewer for raising this critical question. We further clarify the core data construction strategy and contributions of this work. Existing multimodal safety datasets primarily emphasize training models to refuse unsafe content, i.e., generating rejection-style responses when encountering risky inputs. However, they rarely provide data that explicitly guides models in performing safe reasoning. As a result, while models can refuse unsafe requests, it often induces overcautious behavior, causing models to reject benign or complex queries and leading to a significant decline in reasoning performance. To address these issues, we construct two types of novel data centered on safe reasoning rather than safe refusal.
>
> First, we build a policy-guided supervised fine-tuning  dataset. Its core idea is not to provide fixed rejection templates, but to teach the model to explicitly incorporate safety policy clauses during reasoning. Visual inputs, risk categories, and policy content are structured together to guide the reasoning process. Such examples demonstrate step-by-step alignment of visual evidence, risk assessment, and policy references, enabling the model not only to know what should not be answered, but also why it should not be answered and how to perform safe and interpretable multimodal reasoning based on policy. The construction process, including policy framework design, structured multimodal inputs, reasoning trace generation, and quality filtering, is detailed in Section 4 and Appendix A.
>
> Second, we construct a larger preference learning dataset designed to guide reasoning chains themselves, rather than just the final answers. **To our knowledge, this is the first policy-guided multimodal safe reasoning preference dataset**. Its key innovation lies in addressing typical failure modes of reasoning models by generating “negative reasoning trajectories,” including incorrect reasoning chains extracted from real model failures and adversarially generated sequences that appear reasonable but gradually exhibit safety decay. Positive examples correspond to policy-consistent, visually aligned, and stable reasoning chains. Through such paired preference learning, the model not only learns safe conclusions but also how to maintain safety consistency throughout the reasoning process, mitigating the chain-of-thought safety degradation identified in our study.
>
> In summary, we systematically construct training data targeting safe reasoning in multimodal large models, enabling models to learn how to reason safely and interpretably from policy guidance, visual evidence, and reasoning chain structures. We will further emphasize this contribution in the final version.
>
> ### W2: How the paper addresses potential biases introduced by using AI-generated data.
>
> We thank the reviewer for raising this important question regarding potential biases in AI-generated data. We would like to clarify how our work addresses such biases:
>
> 1. **Iterative Manual Review**
>    Due to the large scale of the dataset to be filtered, large-scale human cross-validation was practically infeasible. To address this, we designed a multi-round data augmentation and filtering framework through iterative optimization. In each iteration, the authors manually reviewed 200–300 samples for qualitative evaluation, which was crucial for adjusting the pipeline. After several rounds, the pipeline outputs achieved approximately 90% agreement with our qualitative judgments.
> 2. **Multi-Round Voting Mechanism**
>    To reduce model-induced bias during data generation, we employed a majority-voting strategy. For each data point, the model produces five independent generations. The final annotation is selected as the version appearing most frequently across rounds, improving annotation consistency and robustness.
> 3. **Position Bias Mitigation**
>    - **During Data Generation**: For negative samples, five candidate annotations are generated per response, and their order is randomized within each round to mitigate known positional biases in LLMs. Data points showing high inconsistency across rounds are filtered out.
>    - **During Evaluation**: When comparing winning rates, the order of responses is randomly swapped to reduce LLM preference for specific positions.

---

> ### Author Response · Authors · 2025-11-27
> **Response to Reviewer 4x5E**
>
> ### Q1: Effectiveness on JailbreakV & Image Description Impact
>
> To evaluate the effectiveness of Policy-guided Safety Tuning (PST) under such challenging conditions, we conducted experiments on the JailbreakV-28K dataset using LlamaGuard for assessment. We report the Attack Success Rate (ASR) for LLM transfer attacks and multimodal LLM jailbreak attacks:
>
> |                  | LLM  | MLLM |
> | ---------------- | ---- | ---- |
> | R1-Onevision+PST | 1.42 | 0.00 |
> | LLaMA-CoT+PST    | 2.14 | 0.07 |
>
> The results indicate that PST substantially improves model robustness against both textual and multimodal jailbreak attacks. Even when presented with random or irrelevant images, MLLMs with PST maintain near-zero ASR, suggesting that the policy-guided safety tuning effectively guides the model to ignore irrelevant visual information that could otherwise trigger unsafe responses.
>
> We observed that irrelevant image descriptions have minimal impact on model behavior after PST, as the model learns to prioritize safe reasoning over spurious visual cues. In practice, this means that PST enhances the model’s alignment with safety objectives even under adversarial or noisy multimodal inputs.

---

### Meta-Review · Area_Chair_9wGV · 2026-01-12

**Summary:**

This paper studies an important and timely problem: the safety–reasoning trade-off in multimodal large reasoning models (MLRMs). The authors provide a systematic analysis showing that explicit reasoning (e.g., CoT-style tuning) can amplify harmful behavior, and identify concrete failure modes including visual attention drift, flawed reasoning initiation, and chain-of-thought safety attenuation. To address these issues, the paper proposes Policy-guided Safety Tuning (PST), a two-stage framework combining policy-guided supervised fine-tuning (PST-SFT) and safety reasoning preference optimization (SRPO). Across multiple multimodal safety benchmarks, PST substantially reduces harmful outputs while largely preserving general reasoning performance. Reviewers generally agreed the paper is well written, empirically thorough, and addresses a meaningful gap in multimodal alignment.

**Reviewer Concerns:**

The main reviewer concerns fell into four categories:
1. Novelty and positioning: Several reviewers noted that PST builds on established techniques (SFT + DPO-style preference optimization) and questioned whether the algorithmic novelty is incremental.
2. Causal soundness of the safety–reasoning trade-off: Some reviewers were initially unconvinced that reasoning itself, rather than missing safety-aware data, is the root cause of increased harmful behavior.
3. Error analysis: Reviewers asked whether the identified mechanisms (especially visual attention drift and unsafe reasoning patterns) were quantitatively validated and demonstrably mitigated by PST.
4. Data quality, bias, and reproducibility: Concerns were raised about reliance on AI-generated data, potential bias, jailbreak robustness, and clarity around computational and data-generation costs.

The rebuttal addressed most of these concerns convincingly. The authors added controlled comparisons of the same model with and without CoT, supporting the claimed trade-off; provided quantitative breakdowns of failure modes across datasets, showing consistent reductions after PST; added jailbreak robustness experiments with near-zero ASR; clarified data filtering, voting, and bias mitigation strategies; and disclosed compute and API costs alongside a commitment to release code and data. While the novelty remains primarily problem-driven and data-centric rather than algorithmic, this is clearly acknowledged and, in my view, acceptable given the empirical depth and clarity of insights provided.

**Reviewer Scores:**

- 4x5E: Likely no change (already marginally positive; concerns largely addressed).
- vzoQ: Likely no change or slight increase; main conceptual questions were clarified, but reviewer was already borderline positive.
- c61W: Likely no change; acknowledged clarifications, but original assessment was already near threshold.
- ZsCz: Explicitly increased their score after rebuttal, stating that most concerns were addressed and requesting incorporation of clarifications into the final manuscript.

---

### Decision · Program_Chairs · 2026-01-26

Accept (Poster)